# Is There an Indication for First Line Radiotherapy in Primary CNS Lymphoma?

**DOI:** 10.3390/cancers13112580

**Published:** 2021-05-25

**Authors:** Clemens Seidel, Christine Viehweger, Rolf-Dieter Kortmann

**Affiliations:** Department of Radiation Oncology, University Hospital Leipzig, Stephanstraße 9a, 04103 Leipzig, Germany; christine.viehweger@medizin.uni-leipzig.de (C.V.); rolf-dieter.kortmann@medizin.uni-leipzig.de (R.-D.K.)

**Keywords:** primary CNS lymphoma, radiotherapy, cranial irradiation, whole brain radiotherapy neurocognitive deficits

## Abstract

**Simple Summary:**

First line whole brain radiotherapy (WBRT) alone is a palliative treatment for patients with primary CNS lymphoma (PCNSL) not suitable for intensive multimodal therapy. Consolidating WBRT after chemotherapy with high-dose methotrexate WBRT has activity comparable to autologous stem cell transplantation (ASCT). However, dose-dependent toxicity of WBRT is a relevant limiting factor. With standard dose, WBRT rates of neuro-cognitive decline are high and should be taken into account. As an alternative, after complete response (CR) low-dose consolidating WBRT carries a lower risk for neurotoxicity and is a treatment option in patients not suitable for ASCT. For patients not achieving CR, the best combined treatment potentially involving focal radiotherapy remains to be determined.

**Abstract:**

*Background:* Primary CNS Lymphoma is a rare and severe but potentially curable disease. In the last thirty years treatment has changed significantly. Survival times increased due to high-dose methotrexate-based chemotherapy. With intensive regimens involving autologous stem cell transplantation (ASCT), 4-year survival rates of more than 80% can be reached. However, this treatment regimen is not feasible in all patients, and is associated with some mortality. *Methods:* In this review, current evidence regarding the efficacy and toxicity of radiotherapy in PCNSL shall be summarized and discussed mainly based on data of controlled trials. *Results:* Being the first feasible treatment whole brain radiotherapy (WBRT) was initially used alone, and later as a consolidating treatment after high-dose methotrexate-based chemotherapy. More recently, concerns regarding activity and neurotoxicity of standard dose WBRT limited its use. On the contrary, latest evidence of some phase II trials suggests efficacy of consolidating WBRT is comparable to ASCT. After complete remission reduced dose WBRT appears as a feasible concept with decreased neurotoxicity. Evidence for use of local stereotactic radiotherapy is very limited. *Conclusion:* Radiotherapy has a role in the treatment of PCNSL patients not suitable to ASCT, e.g., as consolidating reduced dose WBRT after complete response. Local stereotactic radiotherapy for residual disease should be examined in future trials.

## 1. Introduction

Within the past decades major developments have dramatically changed the therapeutic landscape of primary CNS lymphoma (P-ZNSL). Initially, whole brain radiotherapy (WBRT) served as the sole treatment with modest efficacy [1]. Later, high-dose chemotherapy with methotrexate (HD-MTX) constituted a major breakthrough towards long-term survival [2]. WBRT as a means of consolidating therapy after HD-MTX used to be standard of care but was questioned due to concerns regarding the survival benefit [3] and detrimental neurocognitive effects [4]. More recently, reduced dose WBRT gained attention because of relevant therapeutic effects with low toxicity [5]

Several, mostly small, trials have dealt with the value of WBRT in the past decades. Within this review current clinical evidence concerning the use of radiotherapy in PCNSL shall be outlined with focus on larger controlled clinical trials.

## 2. Methods

A PubMed research has been performed applying the keywords “PCNSL and radiotherapy”; “Primary CNS-Lymphoma and radiotherapy”; “Primary CNS-Lymphoma and neurotoxicity” with focus on primary or secondary results of controlled clinical trials and large series. Concerning newer radiotherapy techniques such as stereotactic radiotherapy, also smaller series have been included. In addition, recent review articles on use of WBRT were compiled and screened for results of controlled trials in PSCNSL.

## 3. Results of Clinical Trials

Results of literature retrieval are described in chronological order and summarized in Table 1.

### 3.1. Radiation as Monotherapy in First Line Setting—Lessons from the Past

In the early days of treatment, PCNSL WBRT was treatment of choice. From this time, small uncontrolled case series describe a median survival of about 1 year [1,6]. Because of multifocality of disease WBRT and not focal radiotherapy was used. The treatment concept comprises radiotherapy of the entire brain including the meninges down to C2 and the two posterior thirds of the orbits [7]. Inclusion of the posterior parts of the eye was deemed necessary due to the PCNSL propensity to spread to the eye [8].

In an early prospective multicenter study (RTOG 83-15) using WBRT with 40 Gy and 20 Gy boost 41 patients with newly diagnosed PCNSL were recruited. Median OS was approximately 12 months, 1y- and 2y-OS, being 45% and 25%, respectively. A remission was described in about 50% of patients. From this trial, that set a standard at this time, modest activity of WBRT with limited OS effect in particular in elderly patients was delineated [9]. Another prospective trial (North Central Cancer Treatment Group 96-73-51) investigated the combination of WBRT (41.1 Gy plus 9 Gy boost) with high-dose steroids in elderly patients (median age 76 years) with newly diagnosed PCNSL. This trial was stopped due to poor overall survival, possibly due to side effects of the high-dose steroids. Responses were observed in 42% of patients [10]. Overall, WBRT alone or in combination with steroids showed very limited palliative efficacy.

### 3.2. MTX Based Chemotherapy Can Confer Long Term Survival—Is Standard Dose Consolidating WBRT Beneficial?

Initial data from combinations of HD-MTX, i.th. MTX and WBRT showed that this treatment concept is far superior to radiotherapy alone. For example, in a small phase II trial patients receiving only WBRT had a median survival of 21.7 months compared to 42.5 months median OS in patients receiving combined treatment modalities [11]. Two other small trials strengthened the notion of significant survival benefits from combined treatment [12,13].

In a larger single arm phase II trial (RTOG 93-10, *n* = 102 patients), initial treatment with MPV and intrathecal MTX was followed by WBRT (SD 1.8 Gy, ED 45 Gy) and high-dose Ara-C. With this regimen, a median OS of 37 months (50.4 months in patients younger than 60 years, 2y-OS: 64%) was achieved [14].

Due to increasing concerns regarding neurotoxicity, efforts were made to omit WBRT. In the so far largest PZNSL trial (“German PCNSL trial”, randomized phase III), 551 patients were treated after high-dose MTX-based chemotherapy with or without WBRT (SD 1,8 Gy, ED 45 Gy). In the trial arm, without WBRT the median OS of 37 months was not different from OS after combined treatment (32.4 months, HR1.06 95% CI 0.8–1.40, *p* = 0.71) while median PFS appeared longer with WBRT (18.3 months vs. 11.9 months, *p* = 0.14) [3]. The study did not show an indication of an OS-benefit, but the primary pre-defined endpoint of non-inferiority with a lower CI margin of 0.9 was not met. Only 318/551 patients in the trial were treated per protocol. In addition, there were imbalances in the trial population after induction chemotherapy. Before WBRT, less patients (36%) were in CR than in the experimental arm (58%). In patients (*n* = 166) not reaching CR after initial treatment WBRT appeared beneficial (per protocol analysis: OS: 24.3 months vs. 18.6 months, HR 0.74, *p* = 0.10; PFS: 5.6 months vs. 3.0 months, HR 0.6, *p* = 0.004).

In 2016 results of the RTOG 0227 phase I+II trial were published implementing a consolidating treatment of temozolomide together with hyperfractionated WBRT (SD: 2 × 1.2 Gy daily, ED: 36.0 Gy) after high-dose MTX and Rituximab. In 53 patients, impressive results of a 2y-OS of 80.8% and a median survival of 7.5 years was reached [15].

### 3.3. Consolidating WBRT vs. ASCT

More recently, autologous stem cell transplantation (ASCT) was used as a consolidating treatment after HD-MTX in order to improve prognosis and to avoid WBRT. Two prospective phase II trials examined ASCT as consolidation compared to consolidating WBRT.

The international “IELSG32” study (randomized phase II trial, [16]) allocated 118 patients in the second randomization step after different initial chemotherapy regimens to either WBRT with a lowered total dose (SD: 1.8 Gy, ED: 36 Gy, in PR plus boost to tumor bed: SD: 1.8 Gy, ED: 9 Gy) or to ASCT. The 2y-PFSR was not different between the two arms (WBRT: 80%, ASCT: 69%, *p* = 0.17). Likewise, the 2y-OS rate did not differ (WBRT: 85%, ASCT 71% (*p* = 0.12). In the “best” initial chemotherapy arm of HD-MTX + AraC + Rituximab + thiotepa (“MATRIX”), a high 4y-OS of 85% (WBRT) and 83% (ASCT) was reached (*p* = 0.39). The trial was not designed to prove superiority of either arm. Two patients in the ASCT arm died from treatment related toxicity. Interestingly, even in patients with proven cerebrospinal dissemination, good survival results after WBRT were seen. Only mild neurotoxicity was observed (see Section 3.5).

In the second recent phase II trial from France (“Precis Study”, 140 patients < 60 years of age were randomized after two cycles of R-MBVP (Rituximab, MTX, VP16, BCNU) + two cycles of R-AraC (+2x intrathecal AraC if CSF pos.) to receive WBRT (SD: 2 Gy, ED: 40 Gy) or thiotepa, busulfan, CPM with ASCT. The primary endpoint of 2-year progression-free survival rates were 63% (95% CI 49% to 81%) and 87% (95% CI, 77% to 98%) in the WBRT and ASCT arms, respectively [17]. In the per protocol analysis, 11% of patients receiving ASCT (5 patients) died from treatment related toxicity (1 patient in WBRT arm). In the intent to treat analysis, 2y-OS rate was 75% after WBRT and 66% after ASCT. The 4y-OS was 64% and 66%, respectively. Cognitive impairment was observed after WBRT, whereas cognitive functions were preserved or improved after ASCT.

### 3.4. Combination of High-Dose Chemotherapy and Consolidating Reduced Dose Cranial Irradiation—Merging the Best of Two Worlds?

Initially, the concept of reduced dose WBRT (rd-WBRT, SD: 1.8 Gy, ED: 23.4 Gy) has proven efficacy and tolerability in the particularly fragile patient population of children with medulloblastoma [18]. In 2013, Morris et al. published the first results of this strategy in P-CNSL. In a single arm phase II trial 52 patients of all age groups received five to seven cycles of R-MPV followed by consolidation rd-WBRT (SD: 1.8 Gy, ED: 23.4 Gy) and consolidation Ara-C in patients with complete response (60% of all patients). Patients without CR received standard dose WBRT. Median PFS of the rd-WBRT arm was 7.7 years (vs. all: 3.3 years). The 2y-PFS in the rd-WBRT arm was: 77%. Median OS in the rd-WBRT group was not reached, in all patients median OS was 6.6 years [5].

These results prompted a comparative randomized phase II trial (RTOG 1114, NCT01399372), applying the same treatment concept. Results are not fully published and need to be treated with caution. However, interim results showed that the study reached its primary endpoint. Patients were stratified by RPA class and randomized to receive R-MPV-A with rd-WBRT (chemoRT arm) vs. R-MPV-A alone (chemo arm). The primary endpoint was intent-to-treat (ITT) PFS. A total of 91 patients were randomized. Among eligible patients, 43 were enrolled in the chemoRT arm and 44 in the chemo arm. Median age was 66 (chemoRT) and 59 (chemo). Median KPS was 80 for both arms. Response rates following R-MPV were 81% (chemoRT) and 83% (chemo). In the chemoRT arm, 37 patients (86%) received rd-WBRT. After median follow-up of 55 months, the median ITT PFS was 25 months in the chemo arm and not reached in the chemoRT arm (HR 0.51; 95% CI (0.27, 0.95); *p* = 0.015). The 2y-PFS was 54% (chemo) and 78% (chemoRT). Median OS was not reached in either arm, with data still maturing. As per investigators’ assessment, the rate of clinically defined moderate to severe neurotoxicity was not different between the two arms [19].

### 3.5. Neurotoxicity after Radiotherapy for CNS-Lymphoma—“The Brain Is an Unforgiving Organ” [20]

There is ample pre-clinical [21,22] and clinical evidence that whole brain radiotherapy is a potentially neurotoxic treatment. Concurring clinical experience ranges from pediatric brain tumors to brain metastases [23,24]. In PCNSL, patients neurotoxicity has become more evident after combining whole brain radiotherapy with high-dose methotrexate. Symptoms range from mild short-term memory deficits to severe dementia with incontinence and walking difficulties [25]. On brain MRI with delay after treatment, mostly white matter scarring combined with different degrees of brain atrophy can be observed [25].

The RTOG 93-10 phase II trial proved a high efficacy of MPV chemotherapy in combination with intrathecal MTX and consolidating WBRT (45 Gy). However, 15% of patients developed severe neurotoxicity and patients >60 years of age were particularly prone to this side effect [14].

In a small prospective phase 2 trial, 31 patients were treated with one cycle of cyclophosphamide, doxorubicin, vincristine, and dexamethasone (CHOD) and two of carmustine (BCNU), vincristine, cytosine arabinoside, and methotrexate (BVAM), followed by cranial radiotherapy (45 Gy whole brain plus a 10-Gy boost for single lesions). Dementia probably related to treatment occurred in five (62%) of the eight patients (60 years or older), and four of them died without evidence of relapse of PCNSL. Dementia correlated with developing brain atrophy and leukoencephalopathy on serial CT or MR scans [26].

More recently, in a secondary analysis of the German PZNSL-SG01 trial cognitive functioning and global health status were reduced in the early WBRT arm as compared to the no early WBRT arm two years after randomization (*p* = 0.004 and *p* = 0.022, respectively). MMSE testing revealed lower values (*p* = 0.002) in the early WBRT arm. A mixed model analysis of longitudinal data additionally showed differences favoring the no early WBRT arm in 15 of 26 dimensions of QoL [4].

In a large meta-analysis including data from 783 patients > 60 years of age, 276 patients received WBRT with a median dose of 36 Gy. Between 15–30% of patients treated with consolidating WBRT developed clinical signs of neurotoxicity while in the entire patient cohort risk of neurotoxicity was approximately 8%. Use of WBRT was associated with increased survival but with a significant elevated risk of symptomatic neurocognitive decline, adjusted odds ratio 5.23, 95% CI 2.33–11.74 [27].

In the IELSG-32 trial, WBRT with a lowered dose (SD: 1.8 Gy, ED: 36 Gy, boost to 45 Gy) was applied as consolidative therapy in 59 patients after high dose MTX-based chemoimmunotherapy. This treatment was generally well tolerated. In 30 assessable patients, improvement of cognitive improvement was seen directly after end of treatment, while two years after treatment some attention and executive function domains were impaired compared to patients treated with ASCT instead of WBRT [16]. The comparatively mild cognitive deficits were attributed to the lowered WBRT dose of WBRT.

Correa et al. characterized cognitive functions in PCNSL patients achieving long-term remission following reduced dose WBRT (23.4 Gy) or HDC-ASCT. Linear mixed model analyses in each group showed statistically significant improvement from baseline up to year 3 in attention/executive functions, graphomotor speed, and memory; however, there was a decline in attention/executive functions and memory after year 3 in both groups. There were no significant longitudinal group differences in cognitive performance or QoL. It was suggested that not only rdWBRT, but also HDC-ASCT may be associated with mild delayed neurotoxicity in progression-free patients [28].

### 3.6. Local Radiotherapy of Residual Disease after Chemotherapy—A New Option?

For patients with PCNSL who experience relevant residual disease after methotrexate-based chemotherapy, WBRT provides a high chance of response, but prognosis is limited. In a series of 27 patients 75% responded to WBRT but median OS stayed poor with 11 months [29].

A more active local treatment of residual lymphoma with reduced neurotoxicity e.g., with high-precision techniques such as stereotactic radiotherapy appears appealing, although, this challenges the general perception of a diffuse intracerebral disease. However, already a post-hoc analysis of the German PZNSL trial raised the opportunity of benefits of local treatments. Progression-free survival (hazard ratio (HR): 1.39) and overall survival (HR: 1.33) were significantly shorter in biopsied patients compared to patients with subtotal or gross total resections. This difference in outcome was not due to age or KPS. When controlled for the number of lesions, the HR of biopsy vs. subtotal or gross total resection remained unchanged for progression-free survival (HR = 1.37; *p* = 0.009), but was smaller for overall survival (HR = 1.27; *p* = 0.085) [30].

Regarding local radiotherapy, Iwabuchi and colleagues examined the value of conventionally fractionated radiotherapy (median end dose: 54.0 Gy) on macroscopic lesions only in patients with residual 1-2 PCNSL lesions after high-dose MTX in a retrospective series. With this approach, the in-field recurrence rate was 26% and the out-of-field recurrence rate was 15% at 3 years for all 24 patients undergoing focal radiotherapy. CNS-recurrence rates were similar in patients undergoing MTX-based chemotherapy and focal radiotherapy to the rates in those undergoing MTX-based chemotherapy and WBRT [31]. In a retrospective series from South Korea, after a high-dose MTX treatment, patients were treated with rd-WBRT (ED 23.4 Gy) and a local boost or standard dose WBRT (ED > 23.4 Gy) in cases of incomplete remission. The 3y-OS and PFS among these patients (*n* = 45) were 77.8% and 53.3% with rd-WBRT, and 58.3% and 45.8% with standard dose WBRT without significant difference [32] (Table 1).

With regards to radiosurgical or stereotactic radiotherapy as a “local curative” approach, evidence is restricted to retrospective series that indicate that stereotactic radiotherapy can be very active in induction of remission of residual lesions [33,34,35,36,37]. Palmer et al. recently performed a literature review on the use of stereotactic radiotherapy/radiosurgery in refractory or recurrent lesions in PSCNL, 16 small series of which only one was prospective and included a control group were reviewed. Mostly single doses from 12–21 Gy were used. As a result, 183 out of 256 evaluated lesions (69%) responded completely to treatment, and in 13 of 204 of the patients (6%) lesions recurred within the treatment area at last follow-up. Overall, the treatment was well tolerated [38].

## 4. Discussion

Evidence for or against the use of WBRT as consolidating treatment in PCNSL is complex and conflicting. Several reviews have discussed the value of radiotherapy, which were mostly very critical [39,40,41,42,43,44] due to justified concerns regarding relevant neurotoxicity and due to results of the German PCNSL trial that showed no significant overall survival benefit from treatment involving standard dose WBRT [3]. In our opinion this view needs to be revisited after the two more recent smaller randomized phase II trials that showed PFS and OS with consolidating WBRT not significantly different from treatment involving ASCT [16,17]. Results of the latter trials justify the conclusion that WBRT is an effective consolidating treatment in PCSNSL. However, these trials proved again dose-dependent neurotoxic risks which are detrimental and may limit use of WBRT [45,46,47]. These toxic effects need to be balanced against relevant short-term toxicity of ASCT that, in itself, carries some mortality risk [16,17]. For patients in complete remission not suitable to ASCT consolidating reduced dose WBRT (SD 1.8 Gy, ED 23.4 Gy) is a reasonable treatment choice balancing long-term chances and risks of this treatment. In patients without CR the value of WBRT is less well defined. Standard dose WBRT can be used, but in light of current evidence of tolerability of reduced dose WBRT and efficacy of local high-precision radiotherapy, a concept merging high-dose local boost and low dose WBRT appears more promising. This concept needs to be validated in future trials. In addition, it should not remain unnoticed that after intensive combined treatment involving ASCT still many patients die from PCNSL relapses, i.e., this treatment can be further optimized. If and which type of consolidating radiotherapy has potential in this “very multimodal” approach remains to be answered in the future. Current evidence-based guidelines appear desirable to better guide treatment decisions.

## 5. Conclusions

Radiotherapy still has a role in the first line setting of PCNSL, e.g., as reduced dose WBRT for consolidation of responding patients not undergoing PBSCT. Standard dose WBRT should be used with caution due to neurotoxicity. The role of local radiotherapy to residual disease within a combined treatment setting remains to be determined.

## Figures and Tables

**Table 1 cancers-13-02580-t001:** Clinical trials.

Author (Year)	Trial	Patient Number	Inclusion Criteria (Age/KPS)	Chemotherapy	Radiotherapy	Endpoint (PFS/OS)	Remarks/Neurotoxicity
Nelson et al. (1992) RTOG 8315 [9]	Phase II Single-arm	*N* = 41	≥18 y, no HIV, no systemic lymphoma, KPS: ≥40	-	WBRT: SD: 1.8 Gy, ED: 40 Gy + boost tumor: 20 Gy	Median OS: 11.6 m 88% failure in 60 Gy region	Prognostic factors: KPS (*p* = 0.002) sex (*p* = 0.032)
DeAngelis et al. (1992) [11]	Phase II Single arm, Group R: late patients	*N* = 47 A) *N* = 31 R) *N* = 16	no HIV	Group A: 1 × MTX 1 g/m^2^ + 6 × 12 mg MTX i.th., Consolidation after RT: 2 × AraC 3 g/m^2^, Group R: RT alone	Group A and R: WBRT: SD: 2 Gy, ED: 40 Gy + boost: SD: 1.8 Gy, ED: 14.4 Gy	PFS: A: 41 m, R: 10 m OS: A: 42.5 m, R: 21.7 m	In pts with multiple lesions: no boost but WBRT: ED: 50 Gy; dementia in 3/31 patients
O’Brien et al. (2000) [12]	Phase II Single arm	*N* = 46	newly diagnosed PCNSL, only brain—no systemic lymphoma, no HIV, ECOG: 0–3	1 × MTX 1 g/m^2^—if CSF positive: intrathecal AraC as long as CSF positive, and 3 times more	WBRT: SD: 1.8 Gy, ED: 45 Gy + boost to tumor bed: SD: 1.8 Gy, ED: 5.4 Gy	median OS: 33 m 2 y-OS: 62% 2 y-PFS: 65%	>3 lesions: no boost, but ED: 50.4 Gy; if CSF positive: CSI: SD: 1.5 Gy, ED: 35 Gy; severe neurotox in 6 pts.
Bessell et al. (2002) [13]	Phase II, multicenter	*N* = 57 SD) *N* = 31 LD) *N* = 26	no HIV, 70 y ≥ age, only brain—no systemic lymphoma	1 cycle CHOD + 2 cycles BVAM	sd-WBRT: SD: 1.8 Gy, ED: 45 Gy + boost: SD: 2 Gy, ED: 10 Gy; rd-WBRT after CR: SD: 1.8 Gy, ED: 30.6 Gy	3y-risk of relapse: sd-WBRT: 29%; rd-WBRT: 70% 3y-OS: 55% 3y-OS after CR: no significant difference between SD and LD	CR: sd-WBRT: 68% rd-WBRT: 77%; cogn. decline in CR 1 y after rd-WBRT: 0/13; after sd-WBRT: 6/10 ≥ 60 y
Bessell et al. (2001) [25]	Phase II Single arm	*N* = 31	no HIV, only brain—no systemic lymphoma	1 cycle CHOD + 2 cycles BVAM	WBRT: SD: 1.8 Gy, ED: 45 Gy + boost SD: 2 Gy, ED: 10 Gy; if CSF positive: + CSI: SD: 1.4 Gy, ED: 35 Gy	median OS: 38 m 5 y-OS: 31%	age +/− 60 y major predictor; dementia occurred in 5/8 (62%) ≥ 60
DeAngelis et al. (2002) RTOG 9310 [14]	Phase II, multicenter	*N* = 102 *N* (RT) = 82	newly diagnosed PCNSL, only brain—no systemic lymphoma, no HIV KPS: ≥50	5 cycles (MTX 2.5 mg/m^2^, vincristine, procarbazine) + 5 × 12 mg MTX i.th.—consolidation after RT: 2 × HD-AraC 3 g/m^2^	RT-Group: SD: 1.8 Gy, ED: 45 Gy, amendments: if CR, SD: 1.2 Gy 2× daily, ED: 36 Gy, if PR: SD: 1.8 Gy, ED: 45 Gy	median OS: 36.9 m <60: 50.4 m >60: 21.8 m 2 y-OS: 64% median PFS: 24 m <60: 38.8 m >60: 11.1 2 y-PFS: 50%	15% severe neurologic toxicities, chemo: 53% Grade 3–4 toxicity
Thiel et al. (2012) [3]	Phase III, randomized	*N* = 551	newly diagnosed PCNSL, ≥18 y, no systemic lymphoma, life expectancy ≥2 m, no immunosuppression (Incl. HIV), KPS: ≥30	(S): 05/2000–08/2006: HD-MTX (4 g/m^2^), 6 × 14-days-cycles 09/2006-05/2009: HD-MTX (same) + Ifo (1.5 g/m^2^), d: 3–5, 6 cycles (E): no CR: HD-AraC, 4×	Standard (S): SD: 1.8 Gy, ED: 45 Gy WBRT Experimental (E): only chemo	median OS: S: 32.4 m, E: 37.1 m median PFS: S: 18.3 m, E: 11.9 m Initial CR 47% arm before WBRT: 36% arm without WBRT: 58%	Primary endpoint OS (non-inferiority-margin 0.9) was not met, 318 patients treated per protocol; neurotoxicity more common after WBRT 49% vs. 26%;
Morris et al. (2013) [5]	Phase II, single-arm	*N* = 52	newly diagnosed PCNSL, ≥18 y, no HIV, no systemic lymphoma KPS: all	R-MPV: 5–7 cycles Consolidation: AraC after RT	CR: rd-WBRT SD: 1.8 Gy, ED: 23.4 Gy, no boost No CR: sd-WBRT: 1.8 Gy, ED: 45 Gy, no boost	median PFS: rd: 7.7 y vs. all: 3.3 y; 2 y-PFS rd: 77%; median OS: rd: 3 y-OS: 87%, median OS not reached vs. all median OS: 6.6 y	60% achieved initial CR and later rdWBRT; no significant cognitive decline during the follow-up period, except for motor speed (*p* < 0.05).
Glass et al. (2016) RTOG 0227 [15]	Phase (I+) II (Phase I TMZ)	(Ph.I: *N* = 13) Ph. II: *N* = 53	newly diagnosed PCNSL, ≥18 y, only brain—no systemic lymphoma, no HIV	5 cycles (rituximab + MTX (3.5 g/m^2^)) + 2 cycles TMZ, for 5 days (100–200 mg/m^2^) Consolidation: 10 cycles TMZ 200 mg/m^2^, for 5days	hyperfractionated WBRT: SD: 1.2 Gy twice-daily, ED: 36 Gy	2 y-OS: 80.8% 2 y-PFS: 63.6% median OS: 7.5 y median PFS: 5.4 y	Mean improvement in MMSE: 2.1Decline: 3%
Ferreri et al. (2017) IELSG32 [16]	international randomized Phase II	*N* = 219 *N* = 118 (2nd random.: 59:59)	newly diagnosed PCNSL, only brain—no systemic lymphoma, no HIV, 18–70 y, ECOG 0–3	1st random: 4 cycles of: A) MTX 3.5 mg/m^2^ + Ara B) MTX 3.5 mg/m^2^ + AraC + rituximab C) MTX 3.5 mg/m^2^ + AraC + rituximab + thiotepa 2nd random.: (D or E) E) carmustine-thiohepa conditioned ASCT	D) WBRT: SD: 1.8 Gy, ED: 36 Gy PR: + boost: SD: 1.8 Gy, ED: 9 Gy	2 y-PFS: (*p* = 0.17) D): 80%, E): 69% 2 y-OS: (*p* = 0.12) D) 85%, E) 71% 4 y-OS: (*p* = 0.39) MATRix +D) or E): D) 85%, E) 83% CR before D: 54%, total CR after D: 95% CR before E: 53%, total CR after E: 93%	Chemo: MATRix (Group C) “best”: 4 y OS: 80%, 2nd randomization not designed to prove superiority of either arm, 2 deaths after ASCT (infections); mild cognitive decline after WBRT
Houillier et al. (2019) Precis [17]	Phase II, randomized, 2-arm-study	*N* = 140 analysed: *N* = 38 in each arm	newly diagnosed PCNSL, 18–60 y	2 cylces R-MBVP (rituximab, MTX, VP16, BCNU) + 2 cylces R-AraC (+2× intrathecal AraC if CSF pos.)ArmB: thiotepa, busulfan, CPM with ASCT	Arm A: WBRT: SD: 2 Gy, ED: 40 Gy	2 y-PFS: A: 63%, B: 87% 2 y-OS: A: 75%, B: 66% 4 y-OS: A: 64%, B: 66% Initial CR rate: 46%	Treatment related deaths ASCT: 5 pts = 11% of per protocol population, WBRT: 1 pt; more neurotoxicity after WBRT

Abbreviations: SD: single dose, ED: end dose, R: Rituximab, P: Procarbazin, V: Vincristin, M: Methothrexat, B: BCNU, A: AraC: Cytarabin, C: Cyclophosphamide, D: Doxorubicin, PFS: progression free survival, OS: overall survival, EFS: event-free survival, QoL: quality of life, CSF: cerebrospinal fluid, PD: progressive disease, CSI: craniospinal irradiation, rd-WBRT: reduced dose WBRT, sd-WBRT: standard dose WBRT, i.th.: intrathecal with Ommaya reservoir, and ASCT: autologous stem cell.

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
