# Peer review of "Is There an Indication for First Line Radiotherapy in Primary CNS Lymphoma?"

_cancers, 2021, doi:10.3390/cancers13112580_

Round 1

Reviewer 1 Report

Please see the file attached

Author Response

This review is very interesting and very useful in the clinical practice.

It’s also well structured but I have some concerns.

  • Methods are missing: an explanation of the modalities of studies selection is

Response: Thank you. A Methods section describing the retrieval of the studies was inserted.

  • Page 3, line 131-133, Authors cited a work of Morris et al. (reference number 5), in which the Med. OS target was not reached: please can Authors explain this point?

Response: Median follow up time before publication of the manuscript was 5.9 years after this time Med. OS in the rd-WBRT group was not reached.  Manuscript was changed accordingly.

  • I think that the column of neurotoxicity should be added in tables, and the studies citing it should be

        Response: We fully agree that neurotoxicity is a very important issue after WBRT and should be described in more detail. We changed the heading of the last column to remarks/neurotoxicity, If not included in the table yet and if available we added short results of the cited trials concerning neurotoxicity to this columns.

  • In my opinion also in the “Discussion” Authors could debate about

Response: It’s undebated that WBRT causes neuro-toxicity in a dose-dependant fashion.

We agree that it should be discussed more but we still kept it short: 

Paragraph was changed (phrases regarding neurotox in yellow): Several reviews have

discussed the value of radiotherapy mostly very critical [40–45] due to justified concerns

regarding relevant neurotoxicity and due to results of the German PCNSL trial that showed

no significant overall survival benefit from treatment involving standard dose WBRT [3]. In

our opinion this view needs to be revisited after the two more recent smaller randomized

phase II trials that showed PFS- and OS with consolidating WBRT not significantly different

from treatment involving ASCT [16,17].

Results of the latter trials justify the conclusion that WBRT is an effective consolidating

treatment in PCSNSL. However, these trials proved again dose-dependant neurotoxic risks

which are detrimental and may limit use of WBRT [46–48]. These toxic effects need to be

balanced against relevant short-term toxicity of ASCT that in itself carries some mortality

risk [16,17]. For patients in complete remission not suitable to ASCT consolidating reduced

dose WBRT (SD 1.8 Gy, ED 23.4 Gy) is a reasonable choice balancing long-term chances and

risks of this treatment, taking account especially reduced neurotoxicity.

  • I think that in the “Discussion” Authors should briefly refer to

Response: A last sentence has been added to the discussion: “Current evidence-based guidelines appear desirable to better guide treatment decisions”. Also a section Conclusions has been added.

  • In my opinion title should be changed because it’s not clear what “first line radiotherapy”

refers to. It refers to RT alone? RT after chemotherapy? RT of residual disease?

Response: Thank you for this valuable suggestion. In our opinion the title refers to all the cases stated by the reviewer. We have been supposed to write a manuscript with this title. We guess it has been intended to keep the reader questioning the title.

Minor comments:

  • Page 2, line 48 “of radiotherapy” instead “or radiotherapy”.

Response: Done

  • Page 3, after line 97, Authors talk about the ASCT. I think that they could insert a specific paragraph WBRT vs ASCT (paragraph 2.3). If Authors decide to insert this new paragraph they have to renumber the other

Response: Thank you. Paragraph inserted.

  • Page 3, line 113, please add “)” to (“Precis Study”.

Response: Done

  • Page 3, line 118 “… In per protocol patients receiving…” I don’t understand “per”.

Response: per protocol patients refers to the per protocol analysis of the clinical trial- Sentence has been rephrased: In the per protocol analysis 11% of patients receiving ASCT (5 patients) died from treatment related toxicity…

  • Page 4, line 141 please spell KPS. Authors spelled Karnofsky performance status (KPS) in page 5 line

Response: Done

  • Page 5, line 189, in “Correa et al” please add “.” after “al” .

Response: Done

  • Page 5, line 197 please renumber paragraph “Local radiotherapy of residual after chemotherapy – a new option?”: 2.5 instead 4.

Response: Done

  • Page 5, line 2016 please correct “approacht” with “approach”.

Response: Done

  • Page 5, line 223 please correct “weree” with “were”.

                 Response: Done

We would like to thank the reviewer for the very kind and constructive review!

Reviewer 2 Report

Thanks for the opportunity to review this interesting manuscript. Overall this is a well written review article and has is some more scope of improvement. Following are my comments.

Prefer the acronym P-CNSL than P-ZNSL.

Standardize the acronyms used – (eg – months, mon, m)

Line 96 and 97: In 53 patients impressive 96 results of a 2y-OS of 80,8% and a median survival of 7,5years was reached

Comment: change the commas – 80.8% / 7.5 years

Line 125: concept of reduced dose WBRT – Readers would be interested to know the dose of RT that is used as part of standard of care versus reduced dose.

Line 181 – correction to commas

Line 220 – Typo – South corea

Comment: change to south Korea

Line 181: In the IELSG-32 trial a “medium dosed” WBRT (SD: 1,8 Gy, ED: 36 Gy, Boost to 45 Gy)

Comment: the Dose of RT needs clarification. The dose seems incorrect.

Discussion

Line 2 and 3: Evidence for or against the use of WBRT as consolidating treatment in PCNSL is complex 3
and partially conflicting

Comment : can avoid the word ‘partially’

Author Response

Reviewer 2

Comments and Suggestions for Authors

Thanks for the opportunity to review this interesting manuscript. Overall this is a well written review article and has is some more scope of improvement. Following are my comments.

Prefer the acronym P-CNSL than P-ZNSL.

Response: OK, changed accordingly

Standardize the acronyms used – (eg – months, mon, m)

Response: OK, changed accordingly

Line 96 and 97: In 53 patients impressive 96 results of a 2y-OS of 80,8% and a median survival of 7,5years was reached

Comment: change the commas – 80.8% / 7.5 years

Response: OK, changed accordingly

Line 125: concept of reduced dose WBRT – Readers would be interested to know the dose of RT that is used as part of standard of care versus reduced dose.

Response: OK, 1.8 Gy to 23.4 Gy was mentioned as reduced dose concept

Line 181 – correction to commas

Response: Done

Line 220 – Typo – South corea Comment: change to south Korea

Response: Done

Line 181: In the IELSG-32 trial a “medium dosed” WBRT (SD: 1,8 Gy, ED: 36 Gy, Boost to 45 Gy) Comment: the Dose of RT needs clarification. The dose seems incorrect.

Response: Done, Sentences was rephrased:” In the IELSG-32 trial WBRT with a lowered dose (SD: 1.8 Gy, ED: 36 Gy, Boost to 45 Gy) was applied as consolidative therapy in 59 patients after high dose MTX based chemoimmunotherapy.”

Discussion

Line 2 and 3: Evidence for or against the use of WBRT as consolidating treatment in PCNSL is complex 3

and partially conflicting

Comment : can avoid the word ‘partially’

Response: Done, “partially” was removed

We would like to thank the reviewer for the kind and constructive review!

Round 2

Reviewer 1 Report

No comments.